# Failure Modes of Variational Autoencoders and Their Effects on Downstream Tasks

## Abstract

Variational Auto-encoders (VAEs) are deep generative latent variable models that are widely used for a number of downstream tasks. While it has been demonstrated that VAE training can suffer from a number of pathologies, existing literature lacks characterizations of exactly *when* these pathologies occur and *how* they impact down-stream task performance. In this paper we concretely characterize conditions under which VAE training exhibits pathologies and connect these failure modes to undesirable effects on specific downstream tasks, such as learning compressed and disentangled representations, adversarial robustness and semi-supervised learning.

## 1 Introduction

Variational Auto-encoders (VAEs) are deep generative latent variable models that transform simple distributions over a latent space to model complex data distributions Kingma & Welling (2013). They have been used for a wide range of downstream tasks, including: generating realistic looking synthetic data (e.g Pu et al. (2016)), learning compressed representations (e.g. Miao & Blunsom (2016); Gregor et al. (2016); Alemi et al. (2017)), adversarial defense using de-noising (Luo & Pfister, 2018; Ghosh et al., 2018), and, when expert knowledge is available, generating counter-factual data using weak or semi-supervision (e.g. Kingma et al. (2014); Siddharth et al. (2017); Klys et al. (2018)). Variational auto-encoders are widely used by practitioners due to the ease of their implementation and simplicity of their training. In particular, the common choice of mean-field Gaussian (MFG) approximate posteriors for VAEs (MFG-VAE) results an inference procedure that is straight-forward to implement and stable in training.

Unfortunately, a growing body of work has demonstrated that MFG-VAEs suffer from a variety of pathologies, including learning un-informative latent codes (e.g.van den Oord et al. (2017); Kim et al. (2018)) and unrealistic data distributions (e.g. Tomczak & Welling (2017)). When the data consists of images or text, rather than evaluating the model based on metrics alone, we often rely on "gut checks" to make sure that the quality of the latent representations the model learns and the synthetic data (as well as counterfactual data) generated by the model is high (e.g. by reading generate text or inspecting generated images visually (Chen et al., 2018; Klys et al., 2018)). However, as VAEs are increasingly being used in application where the data is numeric, e.g. in medical or financial domains (Pfohl et al., 2019; Joshi et al., 2019; Way & Greene, 2017), these intuitive qualitative checks no longer apply. For example, in many medical applications, the original data features themselves (e.g. biometric reading) are difficult to analyze by human experts in raw form. In these cases, where the application touches human lives and potential model error/pathologies are particularly consequential, we need to have a clear theoretical understanding of the failure modes of our models as well as the potential negative consequences on down-stream tasks.

Recent work (Yacoby et al., 2020) attributes a number of the pathologies of MFG-VAEs to properties of the training objective; in particular, the objective may compromise learning a good generative model in order to learn a good inference model – in other words, the inference model over-regularizes the generative model. While this pathology has been noted in literature (Burda et al., 2016; Zhao et al., 2017; Cremer et al., 2018), no prior work has characterizes the *conditions* under which the MFG-VAE objective compromises learning a good generative model in order to learn a good inference model; moreover, no prior work has related MFG-VAE pathologies with the performance on *downstream tasks*. Rather, existing literature focuses on mitigating the regularizing effect of the inference model on the VAE generative model by using richer variational families (e.g. Kingma et al.

(2016); Nowozin (2018); Luo et al. (2020)). While promising, these methods introduce potentially significant additional computational costs to training, as well as new training issues (e.g. noisy gradients Roeder et al. (2017); Tucker et al. (2018); Rainforth et al. (2019)). As such, it is important to understand precisely when MFG-VAEs exhibit pathologies and when alternative training methods are worth the computational trade-off. In this paper, we characterize the conditions under which MFG-VAEs perform poorly and link these failures to effects on a range of downstream tasks. While we might expect that methods designed to mitigate VAE training pathologies (e.g. methods with richer variational families (Kingma et al., 2016)), will also alleviate the negative downstream effects, we find that this is not always so. Our observations point to reasons for further studying the performance VAE alternatives in these applications. Our contributions are both theoretical and empirical:

I. *When* VAE pathologies occur: (1) We characterize concrete conditions under which learning the inference model will compromise learning the generative model for MFG-VAEs. More problematically, we show that these bad solutions are *globally optimal* for the training objective, the ELBO. (2) We demonstrate that using the ELBO to select the output noise variance and the latent dimension results in biased estimates. (3) We propose synthetic data-sets that trigger these two pathologies and can be used to test future proposed inference methods.

II. *Effects* on tasks: (4) We demonstrate ways in which these pathologies affect key downstream tasks, including learning compressed, disentangled representations, adversarial robustness and semi-supervised learning. In semi-supervised learning, we are the first to document the instance of "functional collapse", in which the data conditionals problematically collapse to the same distribution. (5) Lastly, we show that while the use of richer variational families alleviate VAE pathologies on unsupervised learning tasks, they introduce new ones in the semi-supervised tasks.

These contributions help identify when MFG-VAEs suffice, and when advanced methods are needed.

## 2 RELATED WORK

Existing works that characterize MFG-VAEs pathologies largely focus on relating *local optima* of the training objective to a single pathology: the un-informativeness of the learned latent codes (posterior collapse) (He et al., 2019; Lucas et al., 2019; Dai et al., 2019). In contrast, there has been little work to characterize pathologies at the *global optima* of the MFG-VAE's training objective. Yacoby et al. (2020) show that, when the decoder's capacity is restricted, posterior collapse and the mismatch between aggregated posterior and prior can occur as global optima of the training objective. In contrast, we focus on *global optima* of the MFG-VAE objective in *fully general* settings: with fully flexible generative and inference models, as well as with and without learned observation noise.

Previous works (e.g. Yacoby et al. (2020)) have connected VAE pathologies like posterior collapse to the over-regularizing effect of the variational family on the generative model. However while there are many works that mitigate the over-regularization issue (e.g. Burda et al. (2016); Zhao et al. (2017); Cremer et al. (2018); Shu et al. (2018)), none have given a full characterization of *when* the learned generative model is over-regularized, nor have they related the quality of the learned model to its performance on *down-stream tasks*. In particular, these works have shown that their proposed methods have higher test log-likelihood relative to a MFG-VAEs, but as we show in this paper, high test log-likelihood is not the only property needed for good performance on downstream tasks. Lastly, these works all propose fixes that require a potentially significant computational overhead. For instance, works that use complex variational families, such as normalizing flows (Kingma et al., 2016), require a significant number of parameters to scale (Kingma & Dhariwal, 2018). In the case of the Importance Weighted Autoencoder (IWAE) objective (Burda et al., 2016), which can be interpreted as having a more complex variational family (Cremer et al., 2017), the complexity of the posterior scales with the number of importance samples used. Lastly, works that de-bias existing bounds (Nowozin, 2018; Luo et al., 2020) all require several evaluations of the objective.

Given that MFG-VAEs remain popular today due to the ease of their implementation, speed of training, and their theoretical connections to other dimensionality reduction approaches like probabilistic PCA (Rolinek et al., 2019; Dai et al.; Lucas et al., 2019), it is important to characterize the training pathologies of MFG-VAE, as well as the concrete connections between these pathologies and down-stream tasks. More importantly, this characterization will help clarify for which tasks and datasets a MFG-VAE suffices and for which the computational tradeoffs are worth it.

## 3 BACKGROUND

**Unsupervised VAEs (Kingma & Welling, 2013)** A VAE assumes the following generative process:

$$p(z) = \mathcal{N}(0, I), \quad p_\theta(x|z) = \mathcal{N}(f_\theta(z), \sigma_\epsilon^2 \cdot I) \tag{1}$$

where $x$ in $\mathbb{R}^D$, $z \in \mathbb{R}^K$ is a latent variable and $f_\theta$ is a neural network parametrized by $\theta$. We learn the likelihood parameters $\theta$ while jointly approximating the posterior $p_\theta(z|x)$ with $q_\phi(z|x)$:

$$\max_\theta \mathbb{E}_{p(x)} \left[ \log p_\theta(x) \right] \geq \max_{\theta,\phi} \mathbb{E}_{p(x)} \left[ \mathbb{E}_{q_\phi(z|x)} \left[ \log \frac{p_\theta(x|z)p(z)}{q_\phi(z|x)} \right] \right] = \text{ELBO}(\theta, \phi) \tag{2}$$

where $p(x)$ is the true data distribution, $p_\theta(x)$ is the learned data distribution, and $q_\phi(z|x)$ is a MFG with mean and variance $\mu_\phi(x), \sigma_\phi^2(x)$, parameterized by neural network with parameters $\phi$. The VAE ELBO can alternately be written as a sum of two objectives – the "MLE objective" (MLEO), which maximizes the $p_\theta(x)$, and the "posterior matching objective" (PMO), which encourages variational posteriors to match posteriors of the generative model. That is, we can write:

$$\text{argmin}_{\theta,\phi} - \text{ELBO}(\theta, \phi) = \text{argmin}_{\theta,\phi} (\underbrace{D_{\text{KL}}[p(x)||p_\theta(x)]}_{\text{MLEO}} + \underbrace{\mathbb{E}_{p(x)} \left[ D_{\text{KL}}[q_\phi(z|x)||p_\theta(z|x)] \right]}_{\text{PMO}}) \tag{3}$$

This decomposition allows for a more intuitive interpretation of VAE training and illustrates the tension between approximating the true posteriors and approximating $p(x)$.

**Semi-Supervised VAEs** We extend the VAE model and inference to incorporate partial labels, allowing for some supervision of the latent space dimensions. For this, we use the semi-supervised model introduced by Kingma et al. (2014) as the "M2 model", which assumes the generative process,

$$z \sim \mathcal{N}(0, I), \quad \epsilon \sim \mathcal{N}(0, \sigma_\epsilon^2 \cdot I), \quad y \sim p(y), \quad x|y, z = f_\theta(y, z) + \epsilon \tag{4}$$

where $y$ is observed only a portion of the time. The inference objective for this model is typically written as a sum of three objectives, a lower bound for the likelihood of $M$ labeled observations, a lower bound for the likelihood for $N$ unlabeled observations, and a term encouraging the discriminative powers of the variational posterior:

$$\mathcal{J}(\theta, \phi) = \sum_{n=1}^{N} \mathcal{U}(x_n; \theta, \phi) + \gamma \cdot \sum_{m=1}^{M} \mathcal{L}(x_m, y_m; \theta, \phi) + \alpha \cdot \sum_{m=1}^{M} \log q_\phi(y_m|x_m) \tag{5}$$

where the $\mathcal{U}$ and $\mathcal{L}$ lower bound $p_\theta(x)$ and $p_\theta(x, y)$, respectively (see Appendix A); the last term in the sum is included to explicitly increase discriminative power of the posteriors $q_\phi(y_m|x_m)$ (Kingma et al. (2014) and Siddharth et al. (2017)); $\alpha, \gamma$ controls the relative weights of the last two terms. Note that $\mathcal{J}(\theta, \phi)$ is only a lower bound of the observed data log-likelihood only when $\gamma = 1, \alpha = 0$, but in practice, $\gamma, \alpha$ are tuned as hyper-parameters. Following Kingma et al. (2014), we assume a MFG variational family for each of the unlabeled and labeled objectives.

## 4 WHEN VAEs FAIL: PATHOLOGIES OF THE VAE OBJECTIVE

Our first set of contributions characterizes *when* MFG-VAEs fail. In Section 4.1, we identify two pathological properties of the VAE training objective. In Section 4.2 we introduce new synthetic benchmarks that demonstrate these pathologies and can be used to test future inference algorithms.

### 4.1 FORMALIZING PATHOLOGIES OF THE VAE OBJECTIVE

We fix a set of realizable likelihood functions $\mathcal{F}$, implied by our choice of the generative model network architecture. We assume that $\mathcal{F}$ is expressive enough to contain any smooth function, including the ground truth generating function.

**When VAEs Fail I: The ELBO trades off generative model quality for simple posteriors** Intuitively, global optima of the ELBO correspond to incorrect generative models under two conditions: (1) the true posterior is difficult to approximate by a MFG for a large portion of $x$'s, and (2) there does not exist a likelihood function $f_\theta$ in $\mathcal{F}$ with a simpler posterior that approximates $p(x)$ well.

We formalize these intuitive conditions (1) and (2) in the theorem below. First, recall the decomposition the negative ELBO in Equation 3. In this discussion, we always set $\phi$ to be optimal for our choice of $\theta$. Assuming that $p(x)$ is continuous, then for any $\eta \in \mathbb{R}$, we can decompose the PMO as:

$$
\begin{aligned}
\mathbb{E}_{p(x)}\left[D_{\mathrm{KL}}[q_\phi(z|x)||p_\theta(z|x)]\right] = & \Pr[\mathcal{X}_{\mathrm{Lo}}(\theta)]\,\mathbb{E}_{p(x)|\mathcal{X}_{\mathrm{Lo}}}\left[D_{\mathrm{KL}}[q_\phi(z|x)||p_\theta(z|x)]\right] \\
& + \Pr[\mathcal{X}_{\mathrm{Hi}}(\theta)]\,\mathbb{E}_{p(x)|\mathcal{X}_{\mathrm{Hi}}}\left[D_{\mathrm{KL}}[q_\phi(z|x)||p_\theta(z|x)]\right]
\end{aligned}
\tag{6}
$$

where $D_{\mathrm{KL}}[q_\phi(z|x)||p_\theta(z|x)] \leq \eta$ on $\mathcal{X}_{\mathrm{Lo}}(\theta)$, $D_{\mathrm{KL}}[q_\phi(z|x)||p_\theta(z|x)] > \eta$ on $\mathcal{X}_{\mathrm{Hi}}(\theta)$, with $\mathcal{X}_i(\theta) \subseteq \mathcal{X}$; where $\mathbb{E}_{p(x)|\mathcal{X}_i}$ is the expectation over $p(x)$ restricted to $\mathcal{X}_i(\theta)$ and renormalized, and $\Pr[\mathcal{X}_i]$ is the probability of $\mathcal{X}_i(\theta)$ under $p(x)$. Let us denote the expectation in first term on the right hand side of Equation 6 as $D_{\mathrm{Lo}}(\theta)$ and the expectation in the second term as $D_{\mathrm{Hi}}(\theta)$.

Let $f_{\theta_{\mathrm{GT}}} \in \mathcal{F}$ be the ground truth likelihood function, for which we may assume that the MLE objective (MLEO) term is zero. Now conditions (1) and (2) above may be rewritten more formally as:

**Theorem 1.** *Suppose that there is an $\eta \in \mathbb{R}$ such that $\Pr[\mathcal{X}_{\mathrm{Hi}}(\theta_{\mathrm{GT}})]\,D_{\mathrm{Hi}}(\theta_{\mathrm{GT}})$ is greater than $\Pr[\mathcal{X}_{\mathrm{Lo}}(\theta_{\mathrm{GT}})]\,D_{\mathrm{Lo}}(\theta_{\mathrm{GT}})$. Suppose the following two conditions: (1) [True posterior often difficult] there exist an $f_\theta \in \mathcal{F}$ with $D_{\mathrm{Lo}}(\theta_{\mathrm{GT}}) \geq D_{\mathrm{Lo}}(\theta)$ and*

$$
\Pr[\mathcal{X}_{\mathrm{Hi}}(\theta_{\mathrm{GT}})]\left(D_{\mathrm{Hi}}(\theta_{\mathrm{GT}}) - D_{\mathrm{Lo}}(\theta_{\mathrm{GT}})\right) > \Pr[\mathcal{X}_{\mathrm{Hi}}(\theta)]\,D_{\mathrm{Hi}}(\theta) + D_{KL}[p(x)||p_\theta(x)];
$$

*and (2) [No good, simpler alternative] that for no such $f_\theta \in \mathcal{F}$ is the MLEO $D_{KL}[p(x)||p_\theta(x)]$ equal to zero. Then at the global minima $(\theta^*, \phi^*)$ of the negative ELBO, the MLEO will be non-zero.*

Theorem 1 shows that under conditions (1) and (2) the ELBO can prefer learning likelihood functions $f_\theta$ that reconstruct $p(x)$ poorly, even when learning the ground truth likelihood is possible! While the proof is straightforward (Appendix B.1), it is non-trivial to show that the conditions (1) and (2) are satisfied on actual datasets - we do this in Section 4.2.

**When VAEs Fail II: The ELBO biases learning of the observation noise variance** In practice, the noise variance of the dataset is unknown and it is common to estimate the variance as a hyper-parameter. Here, we show that learning the variance of $\epsilon$ either via hyper-parameter search or via direct optimization of the ELBO can be biased.

**Theorem 2.** *For an observation set of size $N$, we have that*

$$
\underset{\sigma^{(d)}_\epsilon{}^2}{\mathrm{argmin}} - \mathrm{ELBO}(\theta, \phi, \sigma^{(d)}_\epsilon{}^2) = \frac{1}{N}\sum_{n=1}^N \mathbb{E}_{q_\phi(z|x_n)}\left[(x_n^{(d)} - f_\theta(z)^{(d)})^2\right].
\tag{7}
$$

Proof in Appendix B.2. The above theorem shows that the variance $\sigma_\epsilon^2$ that minimizes the negative ELBO depends on the approximate posterior $q_\phi(z|x)$, and thus even when the generative model is set to the ground-truth, the learned $\sigma_\epsilon^2$ will not equal the ground truth noise variance if $q_\phi(z|x_n)$ are poor approximations of the true posterior. Certainly, when the learned generative model does not capture $p(x)$ (say, due to the conditions in Theorem 1), then the learned $\sigma_\epsilon^2$ will again be biased.

***Remark:*** While the pathologies in Theorems 1, 2 seem to depend on our choice of variational family, they are actually artifacts of the ELBO. There are objectives that use MFG families to obtain unbiased estimates of the ground truth generative model and $\sigma_\epsilon^2$ (Finke & Thiery, 2019).

## 4.2 BENCHMARKS TO DEMONSTRATE OF VAE TRAINING PATHOLOGIES

We provide novel benchmarks to demonstrate the pathologies described in Theorems 1 and 2. These benchmarks not only provide intuition for these failure modes, they can also serve as benchmarks for testing future VAE objectives and inference. Finally, while examples here are synthetic in order to provide intuition for general failures on down-stream tasks, in Section 6 we describe how each example corresponds to a class of real datasets on which VAEs can exhibit training pathologies.

**Experimental setup** To show that an observed failure is due to pathologies identified in Theorem 1, we verify that: (A) the learned models have simple posteriors for high mass regions where the ground truth models do not, (B) training with IWAE (complex variational families) results in generally superior generative models and (C) the VAE training cannot be improved meaningfully by methods designed to escape bad local optima, i.e. Lagging Inference Networks (LIN) (He et al., 2019). We

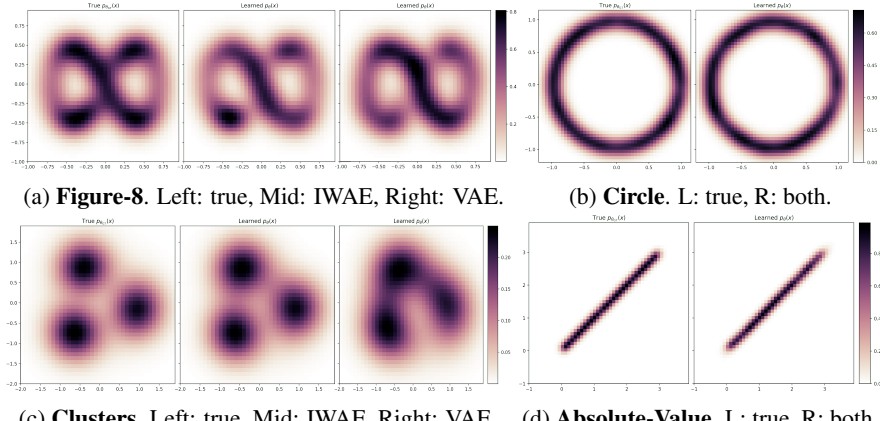

(a) **Figure-8**. Left: true, Mid: IWAE, Right: VAE.    (b) **Circle**. L: true, R: both.

(c) **Clusters**. Left: true, Mid: IWAE, Right: VAE.    (d) **Absolute-Value**. L: true, R: both.

Figure 1: Comparison of true data distributions versus the corresponding learned distributions of VAE and IWAE. When conditions of Theorem 1 are satisfied, in examples (a) and (c), VAE training approximates p(x) poorly and IWAE performs better. When one of the conditions is not met, in examples (b) and (d), then the VAE can learn $p(x)$ as well as IWAE.

specifically chose IWAE as a baseline since we can easily control the complexity of the implied variational family using the number of importance samples $S$, and in doing so we encompass other types of variational families. We ensure that IWAE does not suffer from training issues (e.g. low signal-to-noise ratio) by ensuring $S \leq 20$ and by constructing all toy examples with 1D latent spaces.

We fix a flexible architecture (one that is significantly more expressive than needed to capture $f_{\theta_{\mathrm{GT}}}$) so that our realizable set $\mathcal{F}$ is diverse enough to include likelihoods with simpler posteriors. We train each model to reach global optima as follows: for each method and hyper-parameter settings, we train 5 random initialized randomly, and 5 random with the decoder and encoder initialized to ground truth values. We select the restart with the best value of the objective function. Details in Appendix C.

**Benchmark: approximation of $p(x)$ is poor when Conditions (1) and (2) of Theorem 1 both hold.** Consider the "Figure-8" Example in Figure 1a (described in Appendix F.1). For this dataset, values of $z$ in $[-\infty, -3.0] \cup [3.0, \infty]$ map to similar values of $x$ near $(0, 0)$, where $p(x)$ is high. We verify that, near $x = (0, 0)$, the posteriors $p_{\theta_{\mathrm{GT}}}(z|x)$ are multi-modal, satisfying condition (1) . We verify condition (2) is satisfied by considering all continuous parametrizations of the "Figure-8" curve: any such parametrization will result in a function $f_\theta$ for which distant values of $z$ map to similar values near $(0, 0)$ and thus the posterior matching objective (PMO) will be high. As predicted by Theorem 1, the learned generative model approximates $p(x)$ poorly, learning posteriors that are simpler than those of the ground truth model.

To show that these issues occur because the MFG variational family over-regularizes the generative model, we compare VAE with LIN and IWAE. As expected, IWAE learns $p(x)$ better than LIN, which outperforms the VAE (Figure 1a). Like the VAE, LIN compromises learning the data distribution in order to learn simpler posteriors, since it also uses a MFG variational family. In contrast, IWAE is able to learn more complex posteriors and thus compromises $p(x)$ far less. However, note that with 20 importance samples, IWAE still does not learn $p(x)$ perfectly. See Appendix D.1 for a full quantitative analysis as well as more data-sets for which Theorem 1 holds. Next, we show that *both* conditions of Theorem 1 are necessary for the VAE training pathology to occur.

**Benchmark: approximation of $p(x)$ may be fine when only condition (2) holds.** What happens if the observations with highly non-Gaussian posterior were few in number? Consider the "Circle" Example in Figure 1b (described in Appendix F.2). Here, the regions that have non-Gaussian posteriors are near $x \approx (1.0, 0.0)$, since $z \in [-\infty, -3.0] \cup [3.0, \infty]$ map to points near $(1.0, 0.0)$. However, since the overall number of such points is small, the VAE objective does not trade-off capturing $p(x)$ for easy posterior approximation. Indeed, we see that VAE training is capable of recovering $p(x)$, regardless of whether training was initialized randomly or at the ground truth.

**Benchmark: approximation of $p(x)$ may be fine when only condition (1) holds.** We now study the case where the true posterior has a high PMO for a large portion of $x$'s, but there exist a $f_\theta$ in our

realizable set $\mathcal{F}$ that approximates $p(x)$ well and has simple posteriors. Consider the "Absolute-Value" Example visualized in Figure 1d. Although the posteriors under the ground truth generative model are complex, there is an alternative likelihood $f_\theta(z)$ that models $p(x)$ equally well and has simpler posteriors, and this is the model selected by the VAE objective, regardless of whether training was initialized randomly or at the ground truth. Details in Appendix F.3.

**Benchmark: Theorem 2 implies that the ELBO biases noise variance estimates.** Consider the "Spiral Dots" Example in Appendix F.5. We perform two experiments. In the first, we fix the noise variance ground-truth ($\sigma_\epsilon^2 = 0.01$), we initialize and train $\theta, \phi$ following the experimental setup above, and finally, we recompute $\sigma_\epsilon^2$ that maximizes the ELBO for the learned $\theta, \phi$. In the second experiment, we do the same, but train the ELBO jointly over $\sigma_\epsilon^2$, $\theta$ and $\phi$. Using these two methods of learning the noise, we get $0.014 \pm 0.001$ and $0.020 \pm 0.003$, respectively. The ELBO therefore over-estimates the noise variance by $50\%$ and $100\%$, respectively.

# 5 Impact of Pathologies on Downstream Tasks

In Section 4, we described *when* VAE pathologies occur, both theoretically and through new benchmarks. Now, we describe how how these pathologies can negatively impact specific downstream tasks. On unsupervised tasks, we show that IWAE can avoid the negative effects associated with the MFG variational family over-regularizing the generative model, while LIN cannot. However, surprisingly, IWAE cannot outperform the VAE on our semi-supervised tasks as its complex variational family allows the generative model to overfit.

**Experiment Setup** On unsupervised tasks, we consider only synthetic data, since existing work shows that on real data IWAE learns generative models with higher log data likelihood (Kingma et al., 2016; Cremer et al., 2017). For our semi-supervised tasks, we consider both synthetic data as well as 3 UCI data-sets: Diabetic Retinopathy Debrecen (Antal & Hajdu, 2014), Contraceptive Method Choice (Alcala-Fdez et al., 2010; Dua & Graff, 2017) and the Titanic (Alcala-Fdez et al., 2010; Simonoff, 1997) datasets. In these, we treat the outcome as a partially observed label (observed $10\%$ of the time). These datasets are selected because their classification is hard, and as we will show here, this is the regime in which we expect semi-supervised VAE training to struggle.

## 5.1 Effects on Unsupervised Downstream Tasks

**Disentangled Representations: Failure due to Theorem 1** In disentangled representation learning, we suppose that each dimension of the latent space corresponds to a task-meaningful concept (Ridgeway, 2016; Chen et al., 2018). Our goal is to infer these meaningful ground truth latent dimensions. It's been noted in literature that this inference problem is ill-posed - that is, there are an infinite number of likelihood functions (and hence latent codes) that can capture $p(x)$ equally well (Locatello et al., 2018). In Appendix D.2, we show that, more problematically, the VAE objective can *prefer* learning the representations that *entangles* the ground-truth latent dimensions due to the pathology in Theorem 1.

**Compressed Representations: Failure due to Theorem 1** In practice, if the task does not require a specific latent space dimensionality, $K$, one chooses a $K$ that maximizes the $\log p_\theta(x)$. In Appendix D.3, we show that using a larger $K$ and a smaller $\sigma_\epsilon^2$ means we can capture the data distribution with a simpler function $f_\theta(z)$ and hence get simpler posteriors. That is, increasing $K$ alleviates the need to compromise the generative model in order to improve the inference model and leads to better approximation of $p(x)$. Thus, the ELBO will favor model mismatch ($K$ larger than the ground truth) and prevent us from learning highly compressed representations when they are available. We provide two examples where the ELBO prefers models with larger $K$ over the ground truth model ($K = 1$), and that as $K$ increases, the average informativeness of each latent code decreases, since the latent space learns to generate the observation noise $\epsilon$. We confirm that the posteriors become simpler as $K$ increases, lessening the incentive for the VAE to compromise on approximating $p(x)$. Also, while LIN also shows preference for higher $K$'s, IWAE does not.

**Defenses against adversarial perturbations: Failure due to Theorems 1 and 2** Manifold-based defenses against adversarial attacks (e.g. Jalal et al. (2017); Meng & Chen (2017); Samangouei et al. (2018); Hwang et al. (2019); Jang et al. (2020)) require both accurate estimates of the noise as well as of the intrinsic dimensionality of the data (i.e. the ground truth latent dimensionality); however,

Theorem 2 shows that the ELBO is unable to identify the correct $\sigma_\epsilon^2$ at correct latent dimensionality $K$, and incorrect compression (above, due to Theorem 1) may further result in incorrect noise estimates due to incorrect ground truth latent space dimensionality. See Appendix E for full analysis.

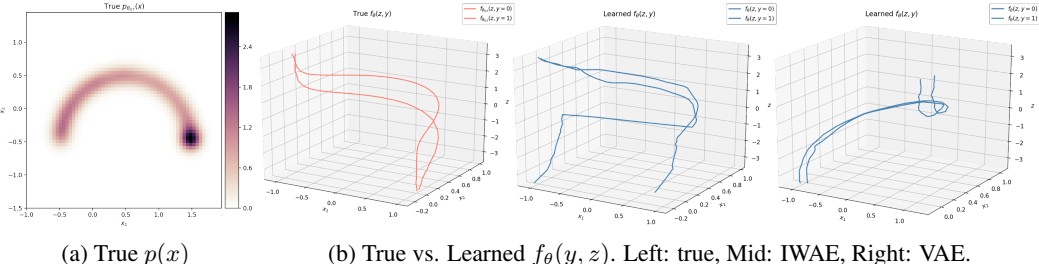

(a) True $p(x)$      (b) True vs. Learned $f_\theta(y, z)$. Left: true, Mid: IWAE, Right: VAE.

Figure 2: **Discrete Semi-Circle**. Comparison of VAE and IWAE for semi-supervision. The ground truth likelihood shows two distinct functions, for $y = 0, 1$. The VAE likelihood is over-regularized by a simple variational family and learns two nearly identical functions ("functional collapse"). The IWAE likelihood function is unregularized and learns two distinct but overfitted functions.

## 5.2 EFFECTS ON SEMI-SUPERVISED DOWNSTREAM TASKS

In semi-supervised VAEs, we assume that some observations $x$ have an additional label $y$ (Kingma et al., 2014). These VAEs have been used for tasks such as generating synthetic cohorts (sampling from $p(x|y = 1), p(x|y = 0)$ respectively), and for generating counterfactuals (generating a synthetic data $x'$ with label $y = 0$ that is similar to a real observation $x$ with $y = 1$). For these tasks, it is important to accurately model the data conditional $p(x|y)$. Surprisingly, Theorem 1 leads to a negative effect specific to semi-supervised tasks, a phenomenon we call "functional collapse": the model ignores the partial labels given by the semi-supervision, causing the learned conditionals to collapse onto a single distribution, $p_\theta(x) \approx p_\theta(x|y = 0) \approx p_\theta(x|y = 1)$. While one might expect methods like IWAE to fix this issue, IWAE actually overfits to the few partial labels due to its rich variational family, causing it to perform no better than a MFG-VAE. In this section, we characterize *when* functional collapse occurs, as well as *how* it impacts semi-supervised downstream tasks.

**Due to Functional Collapse, VAEs trade-off between generating realistic data and realistic counterfactuals** In real datasets, we often have samples from multiple cohorts of the population. General characteristics of the population hold for all cohorts, but each cohort may have different distributions of these characteristics (Klys et al., 2018). Formally, this means that all of the cohorts to lie on a shared manifold but each has a different distribution on that manifold (that is $x|y = 0$ and $x|y = 1$ lie on the same manifold but $p(x|y = 0) \neq p(x|y = 1)$). On such data, the ground truth model's posterior for the unlabeled data $p_{\theta_{\text{GT}}}(z|x) = \int_y p_{\theta_{\text{GT}}}(z, y|x) dy$ will be multi-modal, since for each value of $y$ there are a number of different likely $z$'s, each from a different cohort. As such, using a MFG variational family for the unlabeled portion of the semi-supervised objective ($\mathcal{U}$ in Equation 5) will encourage inference to either compromise learning the data-distribution in order to better approximate the posterior, or to learn the data distribution well but approximate the posterior poorly, depending on our prioritization of the two objectives (indicated by our the choice of the hyperparameter $\gamma$ in Equation 5). In the first case, data generation will be compromised because the model will overfit to the partial labels; however the model will at least be able to generate from two distinct data conditionals $p(x|y = 0)$ and $p(x|y = 1)$. In contrast, in the latter case the learned model will be able to generate realistic data but not realistic cohorts since the model will over over-regularize the likelihood function $f_\theta(z, y)$ to collapse to the same function for all values of $y$ (functional collapse). thereby collapsing the data conditionals $p_\theta(x|y) \approx p(x)$. That is, $p(x|y)$ will generates identical looking cohort regardless of our choice of $y$.

We empirically demonstrate the trade-off between realistic data and realistic counterfactuals generation on the "Discrete Semi-Circle" Example in Figure 2a (full details in Appendix G.1). In this data-set, we show that the MFG-VAE is able to learn the data manifold and distribution well. However, the training objectives learns a model with a simple posterior (in comparison to the true posterior), causing the learned $f_\theta(z, y)$ to collapse to the same function for all values of $y$ (Figure 2b). As a result, $p_\theta(x|y) \approx p_\theta(x)$ under the learned model. As expected, functional collapse occurs

when training with LIN as well. In contrast, IWAE is able to learn two distinct data conditionals $p_\theta(x|y=0), p_\theta(x|y=1)$, but it does so at a cost. ***IWAE does not regularize the generative model, and thus overfits to the few partial labels*** (Figure 2b). Lastly, IWAE learns $p(x)$ considerably worse than the VAE, while learning $p(x|y)$ significantly better. In Appendix D.4, we provide a full quantitative and qualitative analysis of the above on synthetic and real datasets. We also demonstrate an additional pathology for the case where $y$ is continuous, in which a naive choice of the discriminator $q_\phi(y|x)$ causes functional collapse even in IWAE.

## 6 IMPLICATIONS FOR PRACTICE

In this paper, we have presented two sets of contributions that advance our fundamental understanding of VAEs: first, we described the theory of *when* pathologies occur and introduced benchmarks to expose them, second, we described the *impact* of these pathologies on common downstream tasks. Now, we connect these insights with implications for using VAEs in practice.

**How MFG-VAE training pathologies manifest in many real datasets.** Our previous empirical demonstration of the pathology in Theorem 1 are synthetic; here we describe how the conditions of Theorem 1 can manifest in real datasets. The "Figure-8" Example in Figure 1a generalizes to any data manifold with high curvature (e.g. images from videos showing continuous physical transformations of objects), i.e. where the Euclidean distance between two points in a high density region on manifold is (A) less than the length of the geodesic connecting these points and (B) within 2 standard deviation of observation noise. The "Clusters" Example in Figure 1c generalizes to cases where we are learning low-dimensional representations of multimodal data distributions (e.g. popular image datasets where similar images lie in clusters). On these datasets, the VAE training objective would prefer compromising the quality of the generative model for posteriors that are easy to approximate. For the pathology noted in Theorem 2, we expect that the ELBO yields biased estimates of the observation noise whenever the learned model approximates $p(x)$ poorly. Finally, we expect the MFG-VAE's performance on semi-supervised downstream tasks to suffer when a neural network is unable to accurately predict $y|x$. Difficulty in training a good classifier indicates that $x|y=0$ and $x|y=1$ lie roughly on the same manifold, and are difficult to distinguish, leading to the pathologies described in Section 5.2.

**How to make inference choices** We make three simple guidelines for practitioners when using VAEs in order to avoid the pathologies described in this work. While the guidelines are simple, our work provides formal rationales for *why* these practices matter (and we note that these best practices are not always used - e.g. it is common to set $\sigma_\epsilon^2 = 1$ without examining the data-set, or to learn it by optimizing jointly with model parameters (Lucas et al., 2019)). Finally, as others have noted (Finke & Thiery, 2019), a single methodological innovation is unlikely to fix all issues – each innovation makes a specific tradeoff; thus, improvements will need to be task/data specific. Our guidelines are: (1) Set the noise variance $\sigma_\epsilon^2$ using domain expertise, or by hyper-parameter selection with an unbiased low-variance log-likelihood estimator. (2) Investigate the topology of the data (e.g. using topological data analysis, dimensionality reduction) before choosing a variational family. If the data lies on a manifold in distorted Euclidean space (e.g. "Figure-8" Example), or if the data is clustered (e.g. "Clusters" Example), use a rich variational family if you need to learn a very low-dimensional latent space. (3) Whenever using a rich variational family, apply regularization (e.g. $\ell_2$) to the decoder network weights to prevent overfitting. Lastly, (4) on semi-supervised tasks, use a rich variational family for $q_\phi(z|x)$ in the unlabeled data objective ($\mathcal{U}$ in Equation 5) when a simple neural network predicts $y|x$ with low balanced-accuracy (and use a MFG variational family otherwise).

## 7 CONCLUSION

In this work we characterize conditions under which global optima of the MFG-VAE objective exhibit pathologies and connect these failure modes to undesirable effects on specific downstream tasks. We find that while performing inference with richer variational families (which increases training time) can alleviate these issue on unsupervised tasks, the use of complex variational families introduce unexpected new pathologies in semi-supervised settings. Finally, we provide a set of synthetic datasets on which MFG-VAE exhibits pathologies. We hope that these examples contribute to a benchmarking dataset of "edge-cases" to test future VAE models and inference methods.

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
