# OpenReview forum: "Failure Modes of Variational Autoencoders and Their Effects on Downstream Tasks"
_ICLR.cc/2021/Conference — Reject_

### Official Review · AnonReviewer3 · 2020-10-27
**Interesting topic for paper, but serious shortfalls in approach**

**Rating:** 4
**Confidence:** 5

**Review:**

This paper looks to investigate when VAEs fail to learn the maximum marginal likelihood (MML) model and some of the implications this can have for downstream tasks.  In particular, it introduces a theorem (Theorem 1) that provides assumptions under which MML will not be found, and then performs experiments to assess behavior in scenarios where the authors believe these assumptions are satisfied or not.

Though the topic of the paper is interesting and the long term aims are a good line of research, I believe the actual approach taken is rather misguided and that the arguments and conclusions are not properly supported.  In short, I do not believe that the key claimed contribution of describing *when* pathologies occur in VAEs is actually accurate or that the paper adds notable additional insight on this compared to previous work.  As such, I do not believe it is suitable for publication at ICLR in its present state.

*Strengths*
- The problem the work is trying to tackle is important.
- The supplement is very comprehensive and a lot of experiments have been run.
- The paper is generally well written (though the hand-waviness of some of the arguments and reliance on the supplement do significantly detract from the clarity in the latter sections).
- The work is mostly well referenced.  One important missing reference that should be added is  https://arxiv.org/abs/2006.10102 which already makes important related arguments about trade-offs in M2-style semi-supervised VAEs (in particular, they discuss the fact this type of semi-supervision imposes a mismatch between the marginal posterior q(z) and the prior p(z), which closely relates to this paper's discussion of mismatch between p(z|x) and q(z|x) but without needing to assume a ground truth posterior).  Direct discussion about the fact that a VAE need not learn a "ground truth likelihood" to perfectly match the data distribution should also be added (see e.g. http://ruishu.io/2017/01/14/one-bit/).
- Though I am not sure I agree with all the associated conclusions or assumptions being made, it is clear that the authors have put noticeable effort into trying to link the findings of the paper to practical implications.

*Weaknesses*
- I believe the core result of the paper (Theorem 1) is redundant (see below).
- The work makes lots of unnecessary and restrictive assumptions about a "ground truth" model that do not match up to typical real-world situations where such a model does not generally exist (in particular, the latent variables are generally an arbitrary construction such that the concept of a ground truth is actually meaningless).  Moreover, it is already well established that VAEs are not, in general, capable of uncovering a ground truth model even when this does exist (due to equivalence classes amongst other things); the paper offers little beyond this already well-known fact.  In fact, it is written in a way that often implies that we would expect such a ground truth to be uncovered even though various works have explained why this is not a reasonable expectation (e.g. Locatello et al 2019).
- Most of the paper is extremely hand-wavy and imprecise.  For example, various claims are made about the assumptions in Theorem 1 holding or not for different experiments, but the justifications for these are never really properly explained, let alone formally demonstrated.  Furthermore, I actually do not necessarily agree that the assertions made are always correct.  At the very least, it is certainly not the case that all the conclusions of the work have been fully demonstrated.
- The paper constantly implicitly implies that a failure to satisfy the assumptions of Theorem 1 means that the associated pathologies will be avoided.  However, this logic does not hold as the assumptions in the theorem are sufficient not necessary conditions.  Moreover, I show later in the review that actually the result holds under far weaker assumptions, strongly imply that these inverse assertion that the paper relies on are likely to be false.
- The paper is written in a way that incorrectly implies various well-known behaviors are surprising insights (e.g. "the ELBO can prefer learning likelihood functions $f_{\theta}$ that reconstruct $p(x)$ poorly, even when learning the ground truth likelihood is possible!").
- Too many of the experimental results are partially relegated to the appendices.  The paper would be stronger for being more focused on certain aspects of the experiments gone through carefully (with the results themselves actually in the paper!) and some completely relegated to the appendices.  At the moment the paper feels almost unfinished because it is not at all self-contained without the supplement, which is not really acceptable in this format.
- The paper often talks in overly general terms that are not properly justified.  For example, the abstract just talks about generic "pathologies" whereas the topic of the paper is about some quite specific issues rather than a general analysis of different VAE pathologies.
- The figures are scruffy and rather difficult to read (in particular their font size is ridiculously small).

*Redundency of Theorem 1*

I believe that the core result, Theorem 1, is a rather convoluted way of making somewhat simpler points (see below) and does not provide any particular insights; its first assumption is very much reversed engineered rather than a natural starting point.  As such, it is difficult to take intuitions from it or to understand what is required to satisfy it.  Moreover, the experiments seem to demonstrate that the conditions cannot formally be demonstrated in practice as only very hand-wavy explanations are given rather than concrete demonstrations.

Even more problematically, I believe that the Theorem itself is actually a vacuous result in light of simpler, well-known, ideas.  In short, it is obvious and well-known that we will not achieve maximum marginal likelihood (MML) solutions by maximizing the ELBO if the expected KL cannot be driven to zero at the MML value of theta (except in the bizarre edge case where the expected KL does not vary with theta).  It is also straightforward and well-known that this will give a non-zero KL(p(x)||p_{theta}(x)) except in the special case where there is an alternative likelihood that gives the same p_{\theta}(x) while allowing an expected KL of zero (interestingly, this special case may well occur with infinite data though as this allows the encoder variance to tend to zero).   As such, the result of the Theorem is somewhat obvious even without the first assumption holding: this core assumption is much stronger than it needs to be (it is only a sufficient condition, not a necessary one), which not only makes the theorem predominantly redundant, it also undermines most of the subsequent conclusions the paper derives from this result (e.g. the suggestion that pathologies will not occur if this assumption does not hold).

To demonstrate my misgivings more concretely, and verify that the result is indeed vacuous, I developed the following, which I believe to be a stronger and more intuitive alternative result (that also has the key advantage of not making any assumptions about some ground truth model)

*Alternative Theorem*

Consider a VAE with a fixed prior $p(z)$. Define the sets
$$\Theta_{MML} = argmin_{\theta} KL(p(x)||p_{\theta}(x)),$$
and
$$\Theta_{opt} = argmin_{\theta\in\Theta_{MML}} \min_{\phi} E[KL(q_{\phi}(z|x) || p_{\theta}(z|x))],$$
such that $\Theta_{opt} \subseteq \Theta_{MML}$ (note both of these will often be the same single point). If the following assumptions hold:
1. $\nexists \theta \in \Theta_{MML} : E_{p(x)}[KL(q_{\phi}(z|x) || p_{\theta}(z|x))]=0$, i.e. the encoder cannot match the posterior (this assumption is not strictly necessary but is included for intuition because the next assumption cannot hold if this one does not, while it will almost always hold in practice when this assumption does hold),
2. $\exists \theta \in \Theta_{opt} : \lVert \nabla_{\theta} \min_{\phi} E[KL(q_{\phi}(z|x) || p_{\theta}(z|x))] \rVert >0$ and $\nabla_{\theta} \min_{\phi} E[KL(q_{\phi}(z|x) || p_{\theta}(z|x))]$ is locally absolutely continuous for this $\theta$ (see discussion of this assumption below)

then the global minima of the negative ELBO, {$\theta',\phi'$}, is non-optimal for the marginal maximum likelihood in the sense that $\theta'\notin\Theta_{MML}$ and thus:
$$KL(p(x)||p_{\theta'}(x))>KL(p(x)||p_{\theta_{MML}}(x))\ge0$$
where $\theta_{MML}$ be an arbitrary element in $\Theta_{MML}$.

*Proof*:
The proof follows by considering an arbitrary $\theta$ for which the second assumption is satisfied.  Here $\nabla_{\theta} KL(p(x)||p_{\theta}(x))=0$ but $\nabla_{\theta} \min_{\phi} E[KL(q_{\phi}(z|x) || p_{\theta}(z|x))] \neq 0$ so, by our continuity assumption, it must be possible to improve the ELBO by moving $\theta$ in the direction $-\nabla_{\theta} \min_{\phi} E[KL(q_{\phi}(z|x) || p_{\theta}(z|x))]$.  However, because we are improving $\min_{\phi} E[KL(q_{\phi}(z|x) || p_{\theta}(z|x))]$, it impossible for this change to produce a $\theta$ that remains in the set $\Theta_{MML}$  (as this would imply our original point was not in $\Theta_{opt}$).  As we have a $\theta$ that improves the ELBO but is no longer in $\Theta_{MML},$ we can conclude that the optimum of the ELBO is no longer optimal from the perspective of the maximum marginal likelihood. ☐

Here that the second assumption in my Theorem above is very weak whenever the first assumption holds because it effectively equates to saying that the global optima for the MML are not all also local optima for the attainable expected KL divergence between the encoder and posterior.  For example, one would usually not expect the $\theta \in \Theta_{MML}$ to be connected, in which case the assumption can only be violated if the gradient of the attainable expected KL divergence coincidentally happens to be zero for every MML optimal $\theta$.  In light of this, my suggested theorem is making far weaker and (arguably) more intuitive assumptions than Theorem 1 in the paper, while it is also a slightly stronger final result as it does not require there to be a "ground truth" (which is a massive assumption to be able to drop).

Now, of course, the existence of an alternative theorem does not undermine the contributions of the work in itself, but the problem here is that my result above shows that Theorem 1 in the paper is vacuous because its result effectively always holds in practice if the encoder cannot exactly match the posterior (or more precisely, $\nexists \theta \in \Theta_{MML} : E_{p(x)}[KL(q_{\phi}(z|x) || p_{\theta}(z|x))]=0$).  As such, the complex first assumption in Theorem 1 is generally not necessary and offers little insight.  Moreover, because it is obvious that optimizing the ELBO does produce the MML parameters if the encoder can match the posterior, we see that Theorem 1 provides very little insight other than this already well-known fact.  In my opinion, this severely undermines the contribution of the work.

---

> ### Author Response · Authors · 2020-11-13
> **Response to AnonReviewer3**
>
> We thank the reviewer for their thoughtful response, including praise for the paper's overall clarity, comprehensive experiments, and tackling an important problem. We believe that the concerns of the reviewers stems from a few misunderstandings about our goals and claims. We hope to clarify these misunderstanding below:
>
> **Concern: "Strange assumptions" about the ground-truth model, that in real-life doesn't exist. Generally, VAEs cannot recover a ground truth model.**
>
> First, we would like to clarify that Theorem 1 is not concerned with learning the ground truth model $f_\theta(z)$, but rather the true data distribution $p(x)$. As the reviewer notes, it is well known that it is not possible to recover the ground-truth model (e.g. [1, 2]). Our assumptions about the ground-truth model is only: that the data is generated by our generative model (i.e. the decoder is complex enough to capture $p(x)$).  While in real-life we frequently working under model mis-match, for theoretical analysis, it is important to verify that when the model is well-specified, our inference procedure can recover the ground truth distribution $p(x)$. The motivation behind this analysis is consistent with works like [3] and [4].
>
> **Concern: Paper is extremely hand-wavy and imprecise. It is not clear whether the conditions for Theorem 1 holds for various experiments.**
>
> For each example, we check whether or not conditions 1 and 2 of Theorem 1 are satisfied (see Appendix F and G). Specifically:
> 1. We check that the decoder can capture $p(x)$. We generate each example using a ground truth generative model and use the same architecture during inference.
> 2. For assumption 1 of theorem 1, we verify that the posteriors under the ground truth generative models are multimodal or highly skewed for observations that are located in high density regions (Appendix H.2).
> 3. We show that under learned generative model, the posteriors are unimodal and more symmetric (i.e. easily approximated by a MFG) (Appendix H.2).
> 4. We verify that the ELBO is higher for the learned model rather than for the ground-truth model.
> 5. For assumption 2 or theorem 2, we verify that there are no generative models that can capture $p(x)$ but have unimodal and symmetric posteriors.
>
> We will be sure to clarify this in the writing and move this information into a designated section.
>
> **Concern: The paper plays off various well-known behaviors as new and surprising insights.**
>
> In fact, in the introduction, we note that many works have previously commented on the over-regularizing effect of the variational posterior on the learned generative model (e.g. [5, 6, 7, 8]). We also state that what has not been shown are examples where exact conditions triggering this effect are highlighted; furthermore we stated that previous works have not connected this effect to negative consequences on down-stream tasks - these are the goals of our work. As such, we hope the reviewer recognizes the following contributions:
> * We propose a collection of simple toy data-sets that trigger problems with VAE inference, constructed based on our theoretical analysis.
> * We propose a collection of simple toy data-sets on which downstream task performance is compromised.
> * We provide novel documentation and characterization of two pathologies of M2 semi-supervised VAEs: one due to MFG variational family causing functional collapse (Section 5.2), and one due to the added discriminator term (Appendix D.4).
>
>
>
> **Concern: Redundancy of Theorem 1.**
>
> As we noted in the paper, the proof of theorem 1 is obvious - it's a straightforward unfolding of definitions. What's not straight-forward is posing the theorem such that its conditions easily and interpretably allow one to construct "edge-case" data-sets that will trigger the aforementioned pathologies. While we appreciate that the reviewer took the time to propose an alternative theorem, debating the merits of the reviewer's proposed theorem is inappropriate for this context. We would like to note, however, that the reviewer's proposed theorem is not useful for the purposes of our goals in this paper: the reason we chose to frame our Theorem 1 using conditions 1 and 2 is that these conditions give concrete instructions on how to construct datasets that will trigger the over-regularizing effect of the variational posterior (Appendix F and G). The design of these counterexamples leads to insights on failures of VAEs on real datasets and tasks.
>
> References:
> 1. Section 3, Challenging Common Assumptions in the Unsupervised Learning of Disentangled Representations
> 2. Section 6.2, Diagnosing and Enhancing VAE Models
> 3. Frequentist Consistency of Variational Bayes
> 4. Consistency of posterior distributions for neural networks
> 5. Importance Weighted Autoencoders
> 6. Towards Deeper Understanding of Variational Autoencoding Models
> 7. Inference Suboptimality in Variational Autoencoders
> 8. Amortized Inference Regularization

---

### Official Review · AnonReviewer4 · 2020-10-27
**Interesting analysis in the second part**

**Rating:** 5
**Confidence:** 3

**Review:**

Summary:
The paper presents two analysis: (1) Characterization of when the training of VAEs using the ELBO leads to suboptimal generative models (biased towards ones with simple posteriors); and (2) How this suboptimality may affect downstream tasks that use the learned models. Specifically, the work focuses on VAEs using mean-field Gaussians as variational distributions, and explores how their limited modeling capacity affects the final generative model learned. They present some theoretical results and simple and illustrative scenarios. In addition, they present an analysis regarding how the suboptimal models learned affect other tasks, such as learning disentangled or compressed representations.


Pros:
- The paper is very clearly written.
- The theoretical results are novel, although based on well known ideas and maybe not very relevant from a practical perspective (see "cons").
- It presents an interesting exploration of how the known failures of VAEs affect subsequent tasks that use the suboptimal models. I think this is quite relevant, and often does not receive as much attention as new training methods for VAEs.


Cons:
- As mentioned in the paper, the failure modes of VAEs are known. The paper's first contribution is a characterization of when they happen. While the theorems give precise conditions and expressions, it is not clear to me whether they are useful in practice for real scenarios. I think most of the analysis of downstream tasks do not really require the precise conditions given in the theorems, but the ideas behind them, which were already known. For instance, the analysis regarding disentangled representations states that given several models with equal likelihood, optimization will choose the one with lower KL divergence (which may be disentangled or not, depending on the scenario). This does not require the conditions from theorem 1, but knowing that for a given log p(x), decreasing the KL divergence increases the ELBO (which is known, given that the tightness of the ELBO bound is exactly the KL divergence).
- Something similar happens with the analysis regarding compressed representations. Here, it is mentioned that increasing K (dimensionality of latent space) leads to simpler functions and thus to simpler posteriors, and is therefore preferred when using mean-field Gaussians approximating distributions. Again, the precise conditions from the theorems do not seem to be used here. (To the authors, please let me know if I am wrong; that is, if the precise results from the theorems are actually used in these analysis and I missed it.)
- The relevance of the result in theorem 2 is not clear to me. The fact that the optimal noise parameters is not recovered is expected (it is known that optimizing the ELBO leads to biases in the parameters of the generative model), albeit the expression that precisely quantifies the bias in terms of the approximating distribution was not known. However, this exact expression is not used in the subsequent analysis, only the fact that it is biased/suboptimal. Thus the relevance of this result is not clear to me.
- I think that some parts of the second analysis could use further justification. For instance, consider the compressed representations part. It is not clear to me why increasing K leads to generative models with simpler posteriors. I saw the empirical analysis in the Appendix, but did not find an analysis justifying the claim. Am I missing something simple here?


All in all, I do not find the first part of the analysis to be very relevant to the community. This is not the case find the second part of the analysis; I consider that studying how these suboptimalities affect other downstream tasks is important. Despite the fact that I believe the analysis could be expanded, I think it represents a first step in this direction. Therefore, I'm inclined to recommend acceptance. (I updated the score after the discussion.)


One last comment, I would suggest using larger fonts in all plots.

---

> ### Author Response · Authors · 2020-11-13
> **Response to AnonReviewer4**
>
> We thank the reviewer for their thoughtful response, including praise for the paper's clarity, novel theoretical results, and interesting exploration of the consequences of VAE failure modes that often does not receive as much attention to as new VAE inference methods. We address questions and concerns below and will revise the paper to include these points.
>
> **Concern 1: The conditions of Theorem 1 are not used in disentangled representations and compressed representations examples.**
>
> Disentangled representations: Since the toy example in the paper assumes a linear likelihood, the posterior for all $x$'s have the same non-diagonal covariance matrix. Thus, condition 1 of Theorem 1 is satisfied: for all $x$'s, the true posterior cannot be well-captured by a MFG. We left condition 2 unsatisfied to demonstrate that the ELBO prefers learning a model with an entangled latent space but simple posteriors. As the reviewer notes, existing work show that there are infinitely many models that explain $p(x)$ equally well (e.g. [1, 2]). Thus, some works use random restarts to find a model with disentangled representations, selecting that model via human-input [3] or via a new metric [4]. Other works align latent representations with desired concepts with side-information / inductive bias [1, 5, 6]. Our results suggest that these methods will not work well in general, since the ELBO exhibits an inductive bias towards models that entangle the latent representations but have simple posteriors.
>
> Compressed representations: Consider a dataset for which Theorem 1 holds (e.g. the "Clusters" and "Figure-8"). When increasing the latent dimensionality $K$ and decreasing the observation noise variance $\sigma^2_\epsilon$, condition 2 of Theorem 1 no longer holds, since now there exists alternative generative models that explain $p(x)$ well but have simpler posteriors. This happens for two different reasons on the two prototypical data-sets we identify in Section 6. On "Figure-8"-like data, the high $\sigma^2_\epsilon$ causes the posterior for the ground truth model to be multi-modal, an observation $x$ near the crossing of the Figure-8 could have been generated by $z$'s from very different regions in the 1-D latent space (Figure 4a). On the other hand, for a model that captures $p(x)$ equally well but with a smaller $\sigma^2_\epsilon$, the posterior will be less multi-modal (the inverse mapping from x to z will be less ill-posed) and thus be preferred by the ELBO. As the latent dimension $K$ increases, the latent space has more capacity and increasingly models both $f_\theta(z)$ as well as observation noise (the estimated $\sigma^2_\epsilon$ decreases). We observe exactly this phenomenon empirically in Figure 18. On the other hand, to generate the "Clusters"-like data with a 1D latent space, $f_\theta$ contracts regions of the latent space - mapping many different $z$'s to nearby $x$'s (Figure 7b). In this case, the posteriors have high skew and bi-modality (see Figure 7d). By increasing $K$ and decreasing $\sigma^2_\epsilon$, one can learn a $f_\theta(z)$ that becomes more distance preserving. In this case, the posteriors will be unimodal and without skew (see Figure 17), i.e. easily approximated with a MFG.
>
> We will be sure to further emphasize the relationship between the conditions and the downstream tasks in the paper.
>
> **Concern 2: The relevance of Theorem 2 is unclear.**
>
> Theorem 2 is relevant to any task requiring an accurate decomposition between "signal" and "noise" (e.g. adversarial robustness). Concretely, optimizing the ELBO with respect to the noise variance (either by gradient descent or by grid search) will yield biased estimates of the noise and this will bias the estimation of the generative model. That is, learning the wrong noise will cause us to learn the data distribution/manifold incorrectly and/or lead to uninformative latent codes. Although this was observed in existing work [7], it was not previously explained using properties of the ELBO. If one is wondering why use the ELBO to select hyperparameters at all - this is because evaluating $p(x)$ accurately is an open research problem [8, 9], and as such selecting hyperparameters via the ELBO is common practice (e.g. Section 5 of [9]).
>
> References:
> 1. Section 3, Challenging Common Assumptions in the Unsupervised Learning of Disentangled Representations
> 2. Section 6.2, Diagnosing and Enhancing VAE Models
> 3. Interactive Visual Exploration of Latent Space (IVELS) for Peptide Auto-encoder Model Selection
> 4. Unsupervised Model Selection for Variational Disentangled Representation Learning
> 5. Semi-Supervised Learning with Deep Generative Models
> 6. Learning Disentangled Representations with Semi-Supervised Deep Generative Models
> 7. Section 6.2, Don’t Blame the ELBO! A Linear VAE Perspective on Posterior Collapse
> 8. SUMO: Unbiased Estimation of Log Marginal Probability for Latent Variable Models
> 9. Section 3.2, A note on the evaluation of generative models

---

> > ### Comment · AnonReviewer4 · 2020-11-16
> > **Not convinced by the theorems**
> >
> > Thanks for the reply. After reading other reviews and the reply I'd like to add a few things.
> >
> > - I tend to agree with part of the assessment by R3 about Theorem 1, which is in some ways related to my original review. The theorem is stating well-known results (and maybe obvious), albeit in a somewhat formal way with some definitions introduced in the paper. However, I'm still under the impression that the formalism introduced in the Theorem is not adding new insights; and that the effects on other tasks do not use this new formalism either, but the general (and known) ideas behind them (e.g. if posterior is difficult for many x then the "KL penalty" will bias the model parameters learned).
> >
> > - I understand that not being able to recover the true noise variance may be problematic. My point was related to the fact that optimizing the ELBO leads to bias in the model parameters (as mentioned above this is known), and the noise variance is a model parameter. There's no reason to think that this parameter can be learned without bias by optimizing the ELBO. What I'd consider novel is the exact expression for the noise variance. But again, I struggle to see the relevance of the result. In the adversarial attacks analysis this expression is not used, just the fact that the result is biased.
> >
> > I think that  analyzing the effects on other tasks is interesting. But I'm still under the impression that the first part of the paper with the two theorems is not relevant to the community (for the reasons stated above). I'd be happy to keep my acceptance recommendation if I were convinced otherwise. However, after reading other reviews, the replies, and discussing with other reviewers my concerns about this were not dissipated and actually increased. I'll update my score.

---

### Official Review · AnonReviewer2 · 2020-10-28
**Interesting paper that exposes the MLE and posterior matching issues of VAEs**

**Rating:** 5
**Confidence:** 3

**Review:**

This paper exposes the pathologies of VAEs and characterizes them with concrete conditions. The synthetic data experiments well summarize the conditions that we might meet in real-world applications. The authors also analyze the corresponding effects for specific downstream tasks and give insightful suggestions to avoid these problems.

The trade-off between the generative distribution and inference distribution in VAEs has been studied for a long time and has been revealed from several perspectives such as information theory, etc. It is good to see in this paper that the two conditions in theorem 1 summarize well why VAEs work well or poorly.

Questions:
For theorem 1, when only condition 1 holds, we know $p(x)$ might still be approximated but we may get an unwanted generative distribution since there exist several generative models that fit the data equally well. Is there any solution to avoid the duplicated solution?

In the paper, IWAE is applied to avoid the issues mentioned in VAEs. However, in Figure 9, the learned generative model seems to be different than the ground truth, and the posterior is thus simpler than the true posterior. Is there any possible explanation for this multi-modal case?

---

In general, this is an interesting paper that well analyzes the training issues in VAEs and provides insightful guidance to the VAE studies. The paper is well written (the figure labels are too small to read) and easy to follow.


----
Update after the discussion stage

I appreciate the authors' responses to address my questions in the experiments. However, I agree with the concerns of the other reviewers, especially the redundancy of Theorem 1 raised by Reviewer3. After reading the reviews and the discussion, the authors seem not to provide convincing responses to this part and this raises my concerns for this paper. Thus I change my rating below the borderline.

---

> ### Author Response · Authors · 2020-11-13
> **Response to AnonReviewer2**
>
> We thank the reviewer for their thoughtful response, including praise for the paper's clarity, the novelty of identifying *when* pathologies of Theorem 1 appear, and our extensive synthetic experiments.  We address questions and concerns below.
>
> **Question 1:** When only condition 1 of Theorem 1 holds, $p(x)$ will be approximated well--whether by the ground truth solution or some equally-good-at-approximating $p(x)$ decoder function $f_\theta(z)$. The reviewer asks: if an application requires specific desiderata from the latent space, is there a way to select a model with that desiderata (as opposed to some other model that explains $p(x)$ well)?
>
> Among solutions with equal ability to approximate $p(x)$, if we wanted a specific solution, then additional information is needed (e.g. labels from semi-supervision, an informative prior over the function class $f$, etc.). Previous work advocates that with random restarts, one can hopefully find a model with the correct disentangled representation, and then select that model via human-input [1] or via a new metric [2]. Alternatively, previous work argues that one can align the latent representation with the desired latent concepts with side-information / inductive bias [3, 4]. In contrast to previous work, we show that these methods will still not work well -- we show that that the ELBO in fact already exhibits a specific inductive bias for a model that entangles the latent representations but has simple posteriors.
>
> **Question 2:** In Figure 9, as the reviewer points out, IWAE happened to learn a model different from the ground-truth that has uni-modal posteriors (as opposed to the bi-modal posteriors of the ground-truth model). Why did it not learn the ground-truth model?
>
> IWAE's variational family with 20 importance samples is still not a fully-flexible variational family and thus still prefers learning models with simpler posterior. However, since IWAE's variational family is still significantly more flexible than that of a MFG-VAE, it is still able to learn models with more flexible posteriors. In this particular case, instead of learning a bi-modal posterior, it learned a model with a highly skewed posterior, and was thus able to learn $p(x)$ significantly better. We note that on many of the other data-sets, IWAE does learn a bi-modal posterior (Appendix H.2 and H.3).
>
> References:
> 1. "Interactive Visual Exploration of Latent Space (IVELS) for Peptide Auto-encoder Model Selection"
> 2. "Unsupervised Model Selection for Variational Disentangled Representation Learning"
> 3. Section 3 of "Challenging Common Assumptions in the Unsupervised Learning of Disentangled Representations"
> 4. "Learning Disentangled Representations with Semi-Supervised Deep Generative Models"

---

### Official Review · AnonReviewer1 · 2020-10-28

**Rating:** 5
**Confidence:** 3

**Review:**

Summary:
The paper presents a characterization of failure modes of Gaussian VAEs. It is known that Gaussian VAEs can fail to produce good models either by failing to match the data distribution or by learning latent variables that are uninformative. The paper builds upon prior work that suggests that the VAE objective can cause the inference model to over-regularize the generative model. The paper characterizes the conditions under which this over-regularization occurs with corresponding. Furthermore, the paper examines the affect of VAE pathologies on downstream tasks a number of unsupervised and supervised downstream tasks.

----------------------------

The paper is clearly written and the authors have attempted cover all cases of the theoretical reasons they have identified for VAE failures with corresponding experiments. The attempt to cover a number of downstream tasks is also a positive since this is a main goal of unsupervised learning.

As for the core of the paper the authors have identified two types of VAE failures: 1) when the generative quality is traded off for simple posteriors and 2) when the learning of output variance is biased. The first case is presented as occuring when the true posterior is difficult to estimate and there is no good likelhood function with a simple posterior. An expression of this is given as theorem 1.

This brings me to my main concern: Theorem 1 seems to say that if 1) the model cannot match the true posterior (Gaussian assumption is false) and 2) the model cannot match p(x) without matching the true posterior (decoder is not too powerful) then the model will not match p(x). If this is the case, then I would suggest that the statement does not present any new insight into VAE failures since a good VAE model of p(x) would either match the true simple posterior or it will model p(x) without matching the posterior.

Another concern is that although the authors mention prior work on posterior collapse, they do not consider this when characterizing the conditions that cause pathological behvaviour in VAEs. In particular it seems that the "No good, simpler alternative" condition explicitly excludes the case of a strong decoder that is able to ignore the latent code to model p(x) directly. This seems to be at odds with the authors' claim that they consider "fully flexible generative and inference models" when comparing with the prior work of Yacoby et al. Also, the case of strong decoder able to ignore the latent code is difficult to see as a case of the inference model over-regularizing the generative model.

If my understanding above is correct, then I suggest that the authors revise their claim of considering fully flexible models. In particular, they should state whether or not the case of posterior collapse is handled by their characterization.

The authors also present theorem 2 which suggests that using the ELBO to choose output variance results in a biased estimate. As far as I can tell this is new and of interest.

The authors present a number of experiments to show VAE failures: when the true posterior is far from Gaussian, when the true posterior is Gaussian and the decoder is weak, when the posterior is far from Gaussian and but the decoder is strong enough. They also show experiments to show that the output noise is overestimated when the ELBO is used to compute it. One limitation is that the authors have chosen very simple datasets with 1-d latent spaces for illustration. This leads to another question of interest of whether there are problems with VAEs that only appear with higher dimensional latent spaces?

Overall I found the paper to be interesting but in light of the questions mentioned above I do not recommend acceptance.

---

> ### Author Response · Authors · 2020-11-13
> **Response to AnonReviewer1**
>
> We thank the reviewer for their thoughtful response, including praise for the paper's clarity, breadth/coverage of downstream tasks, and the significance of Theorem 2.  We address questions and concerns below.
>
> **Question: If existing literature already shows that the choice of variational family can over-regularize the generative model (leading to poor estimation of $p(x)$), what is the novelty?**
>
> The novelty:
> (a) We describe the conditions under which this over-regularization occurs, which has not been discussed in prior work. While the conditions of Theorem 1 are intuitive, we are the first to theoretically (Appendix B) and empirically demonstrate (Section 4.2) that they are both necessary.
> (b) A collection of simple toy data-sets that trigger the various pathologies, constructed based on these conditions (Appendix F and G): $p(x)$ is approximated poorly ("Clusters" and "Figure-8" examples), biased estimation of the observation noise ("Spiral-Dots" example). We emphasize that existing work on VAE-inference seldom benchmarks proposed methods on simple toy data that exhibit issues that we noted. We hope that these data-sets will be used by researchers to test their inference, and the conditions under which the theorems hold will be used to construct more "edge-case" data-sets.
> (c) Similarly, we share a collection of simple toy data-sets on which downstream task performance is compromised (Appendix F and G): learning compressed representation ("Clusters" and "Figure-8" examples, projected into 5D), cohort / counterfactual generation via semi-supervision ("Discrete/Continuous Semi-Circle" example).
> (d) We provide novel documentation and characterization of two pathologies of M2 semi-supervised VAEs: one due to MFG variational family causing functional collapse (Section 5.2), and one due to the added discriminator term (Appendix D.4).
>
> **Question: If the authors claim to consider "fully flexible generative and inference models", why do they not consider posterior collapse, which occurs when the decoder is powerful enough to ignore the latent code?**
>
> The reason we don't consider this case is because here we only focus on the global optima of the ELBO, and posterior collapse is largely a local optimum (extensively studied, e.g. in [1, 2]). As a global optima, posterior collapse occurs under very restricted conditions: when true posterior equals the prior and is thus perfectly modeled by a MFG variational family $p_\theta(z | x) = q_\phi(z | x) = p(z)$ (e.g. [1, 2, 3]). However, for this condition to occur as a global optima -- that is, for $p_\theta(z | x) = p(z)$ -- the likelihood function must ignore the $z$'s completely $f_\theta(z) = g(\theta)$. Since $\theta$ is not a random variable, this implies that for any non-Gaussian $p(x)$, the observation noise $\epsilon$ needs to be sufficiently flexible to explain the data (and thus must be non-Gaussian). Since this is such a restrictive setting, we did not consider it in the paper.
>
> That said, if the reviewer meant partial posterior collapse - when the latent codes of the learned model are less informative than the latent codes of the ground-truth model - then we *do* cover that. Specifically, in the section on compressed representations, we show that by increasing the latent-space dimensionality and and decreasing the observation noise, the average mutual information of the latent codes decreases (Appendix D.3 and Table 2).
>
> **Clarification: The reviewer says that the paper shows VAE failures for the following cases: "when the true posterior is Gaussian and the decoder is weak, when the posterior is far from Gaussian and but the decoder is strong enough".**
>
> We emphasize that in all of our experiments, the decoder $f_\theta(z)$ is parameterized be a DNN significantly more flexible than needed for any parametrization of the data manifold/distribution (and is thus "*strong*"). In doing so, we (a) avoid the trivial case in which the decoder is unable to model $p(x)$ because it simply does not have enough capacity, and (b) we simulate the conditions under which VAEs are often trained, with very powerful decoders (e.g. for image/NLP-type data).
>
> **Concern: the authors present 1d data-sets to illustrate the problems and wonders whether other issues will occur in higher dimensions**
>
> We specifically focus on 1d examples for several reasons: (a) to show (perhaps surprisingly) how VAEs with flexible decoder and encoder networks can fail drastically even in such simple settings (making these good benchmarks!) and (b) to ensure that none of our empirical claims are caused by difficulties of training that would be exacerbated in more dimensions.
>
> References:
> 1. Section 2.2, "Lagging Inference Networks and Posterior Collapse in Variational Autoencoders"
> 2. "The Usual Suspects? Reassessing Blame for VAE Posterior Collapse"
> 3. Section 4, "Towards a Deeper Understanding of Variational Autoencoding Models"

---

> > ### Comment · AnonReviewer1 · 2020-11-21
> > **Response**
> >
> > The main problem remains that the core of the paper, theorem 1 presents no new insight into VAE failures and the conclusions seem almost trivial. The authors' response does not appear to present anything that would change this evaluation.

---

### Decision · Program_Chairs · 2021-01-07
**Final Decision**

**Decision:**

Reject

**Comment:**

This paper investigates various pathologies that occur when training VAE models. There was quite a bit of discussion (including "private" discussion between the reviewers) about the theory presented. Particular concerns included: For Theorem 1, while the required conditions formalise the setting in which the learned likelihoods are poor, it's unclear whether these particular conditions they are useful in practice or provided deep insight; for Theorem 2, its relevance and importance was not necessarily clear. In general the results in these two theorems are closely related to known challenges (e.g. that using the ELBO to optimise parameters may lead to bias), without necessarily providing as much new insight as one might hope.

I would note that all the reviewers included positive feedback as to the quality of the experiments, showing the impact of these pathologies on downstream tasks. However, as written much of the paper focuses on the theory — too many of the (very interesting!) figures and experimental results are relegated to the appendix.